# Roles of Zinc Finger Protein 423 in Proliferation and Invasion of Cholangiocarcinoma through Oxidative Stress

**DOI:** 10.3390/biom9070263

**Published:** 2019-07-07

**Authors:** Timpika Chaiprasert, Napat Armartmuntree, Anchalee Techasen, Chadamas Sakonsinsiri, Somchai Pinlaor, Piti Ungarreevittaya, Narong Khuntikeo, Nisana Namwat, Raynoo Thanan

**Affiliations:** 1Department of Biochemistry, Faculty of Medicine, Khon Kaen University, Khon Kaen 40002, Thailand; 2Cholangiocarcinoma Research Institute, Khon Kaen University, Khon Kaen 40002, Thailand; 3Faculty of Associated Medical Sciences, Khon Kaen University, Khon Kaen 40002, Thailand; 4Department of Parasitology, Faculty of Medicine, Khon Kaen University, Khon Kaen 40002, Thailand; 5Department of Pathology, Faculty of Medicine, Khon Kaen University, Khon Kaen 40002, Thailand; 6Department of Surgery, Faculty of Medicine, Khon Kaen University, Khon Kaen 40002, Thailand

**Keywords:** zinc finger protein 423 (ZNF423), oxidative stress, cholangiocarcinoma, cancer progression

## Abstract

Zinc finger protein 423 (ZNF423) is a transcriptional factor involved in the development and progression of cancers but has not yet been examined in cholangiocarcinoma (CCA), an oxidative stress-driven cancer of biliary epithelium. In this study, we hypothesized that oxidative stress mediated ZNF423 expression regulates its downstream genes resulting in CCA genesis. ZNF423 protein expression patterns and 8-oxodG (an oxidative stress marker) formation in CCA tissues were investigated using immunohistochemical analysis. The results showed that ZNF423 was overexpressed in CCA cells compared to normal bile duct cells adjacent of the tumor. Notably, ZNF423 expression was positively correlated with 8-oxodG formation. Moreover, ZNF423 expression in an immortalized cholangiocyte cell line (MMNK1) was increased by hydrogen peroxide-treatment, suggesting that oxidative stress induces ZNF423 expression. To investigate the roles of ZNF423 in CCA progression, ZNF423 mRNA was silenced using specific siRNA in CCA cell lines, KKU-100 and KKU-213. Silencing of ZNF423 significantly inhibits cell proliferation and invasion of both CCA cell lines. Taking all these results together, the present study denoted that *ZNF423* is an oxidative stress-responsive gene with an oncogenic property contributing to the regulation of CCA genesis.

## 1. Introduction

Zinc finger protein 423 (ZNF423) is a transcription factor belonging to the Krüppel-like C_2_H_2_ zinc finger protein family. This protein is involved in the regulation of various developmental pathways, especially neurogenesis and adipogenesis [1] through DNA-protein and protein-protein interactions [2]. ZNF423 has previously been reported to cause neurogenesis by activating early B-cell factor 1 (EBF1) and Notch pathways [3,4,5]. In addition, ZNF423 could promote adipogenesis via the peroxisome proliferator-activated receptor γ (PPARγ) pathway [6,7]. Besides, this protein has been identified as an important transcriptional modulator in several types of cancer, such as acute lymphoblastic leukemia (ALL) [8], chronic myelogenous leukemia (CML) [9], nasopharyngeal carcinoma (NPC) [10], neuroblastoma (NB) [11], and breast cancer (BC) [12,13]. In ALL, ZNF423 inhibits transcriptional factor of B-cell differentiation (EBF1) leading to B-cell maturation arrest [8]. Furthermore, in CML, this gene enhances CML cell growth by cooperating with p210BCR/ABL [9]. In the case of NPC, *UBR5-ZNF423* fusion gene which is a hybrid gene formed from exon 1 of *UBR5* and exons 7–9 of *ZNF423* (EBFs binding domain), is an oncogene that drives proliferation and colony forming ability via suppression EBFs activities [10]. On the other hand, ZNF423 acts as a tumor suppressor in both NB and BC [11,12,13]. In NB, ZNF423 interacts with retinoic receptors resulting in the inhibition of cell growth [11]. In BC, ZNF423 is related to estrogen response and induces *BRCA1* tumor suppressor gene expression leading to the induction and repair of double strand breaks [12,13]. ZNF423 plays different roles in various cancer cell types depending on its downstream interactions, tissues specific contents and the multiple functional domains (30 domains). Therefore, the roles of ZNF423 in other cancers e.g., EBFs-, retinoic acid-, and estrogen-related cancer developments should be investigated to gain further basic knowledge and improve treatment outcome for these diseases.

Cholangiocarcinoma (CCA) is a fatal malignancy of bile duct epithelial cells. Recently, EBF1 down-regulation was investigated in CCA tissues and cell lines and found inhibited the tumor progression [14]. Moreover, CYP19A1, an estrogen producing enzyme, was over-expressed and to stimulate estrogen response in the cancer cells [15]. In addition, retinoic acid receptors (RARγ and RXRα) were over-expressed and played roles in CCA progression [16,17]. Besides, CCA has the highest incidence in Northeast Thailand [18,19]. The major risk factor in this endemic area is the infection with liver fluke, *Opisthorchis viverrini* (Ov) [20]. Ov-infection causes chronic inflammation with the overproduction of reactive oxygen species (ROS) and reactive nitrogen species (RNS) resulting in persistent oxidative stress [21,22,23]. Previous studies suggest that CCA is EBFs-, retinoic acid-, estrogen-, and oxidative stress-related cancer.

Oxidative stress could alter expression profiles of oncogenes and tumor suppressor genes leading to cancer promotion/progression of many tumor types including CCA [24,25]. However, not much attention has been paid to the expression levels of oxidative stress-responsive genes in CCA progression. Likewise, ZNF423 plays various roles depending on its downstream interactions in each cell type. Contrastingly, the role of ZNF423 in CCA genesis has not been proved. Therefore, we aimed to evaluate the roles of ZNF423 in CCA genesis in relation to oxidative stress. Firstly, the expression of ZNF423 and the formation of 8-oxo-2′-deoxyguanosine (8-oxodG), an oxidative stress marker, were detected in CCA tissues. Then, the correlation of ZNF423 and 8-oxodG was analyzed in CCA tissues. Furthermore, hydrogen peroxide (H_2_O_2_) treatment was used for induction of *ZNF423* expression in the immortalized cholangiocyte cell line (MMNK1). Finally, the functional roles of ZNF423 in CCA progression were investigated using ZNF423-knockdown CCA cell lines.

## 2. Materials and Methods

### 2.1. Paraffin-Embedded CCA Tissues

Formalin-fixed and paraffin-embedded CCA tissues (*n* = 75) were recruited from the Cholangiocarcinoma Research Institute, Khon Kaen University, Thailand. The protocol of tissue collection was approved by the Ethics Committee for Human Research, Khon Kaen University (#HE571283; date of approval 9th March 2018 and #HE611577; date of approval 18th December 2018).

### 2.2. Human Cell Lines

CCA cell lines, KKU-100 and KKU-213, were obtained from the Cholangiocarcinoma Research Institute, Khon Kaen University, Thailand. The CCA cell lines were also stocked at the Japanese Collection of Research Bioresources (JCRB) Cell Bank (JCRB1568; KKU-100 and JCRB1557; KKU-213). MMNK1 was established and characterized at Okayama University, Japan [26]. KKU-100, KKU-213 and MMNK1 cell lines were cultured in Ham’s F-12 (Life technologies, Grand Island, NY, USA) supplemented with 10% heat-inactivated fetal bovine serum, 100 U/mL penicillin and 100 µg/mL streptomycin (Life technologies, Grand Island, NY, USA) in an incubator with 5% CO_2_ and 95% relative humidity at 37 °C. The medium was changed once every two to three days.

### 2.3. Immunohistochemistry (IHC)

The ZNF423 expression and 8-oxodG formation in CCA tissues were determined using IHC according to the protocol described previously [14]. Rabbit anti-ZNF423 antibody (1:100; Merck KGaA, Darmstadt, Germany) and mouse anti-8-oxodG antibody (1:200; Japan Institute for the Control of Aging, Shizuoka, Japan) were used as primary antibodies. The peroxidase-conjugated Envision™ anti-rabbit or anti-mouse antibodies (DAKO, Glostrup, Denmark) were used for secondary antibodies. The color was developed using a DAB substrate kit (Vector Laboratories, Inc., Burlingame, CA, USA). The stained areas were analyzed under a light microscope. Immuno-reactivities were assessed by calculating the total immunostaining index (IHC score) as the product of a frequency and intensity scores in each slice. The frequency score describes the estimated fraction of positively stained cancer cells (0 = none; 1 = 1−25%; 2 = 26−50%; 3 = 51−75%; 4 > 75%). The intensity score represents the estimated staining intensity (0 = negative staining; 1 = weak; 2 = moderate; 3 = strong). The IHC score for each sample was obtained by multiplying the frequency and intensity scores, yielding a range of the IHC score 0 to 12. In this study, low and high expressions were arbitrarily determined using the mean of IHC scores of CCA tissues as the cut-off value.

### 2.4. RNA Extraction from CCA Cell Lines and Reverse Transcription

Total RNA was extracted from approximately 1 × 10^6^ cells using TRIZOL^®^ reagent (Life technologies, Carlsbad, CA, USA). The RNA solution was kept in −80 °C until used. RNA samples were reversed into complementary DNA (cDNA) using a High-capacity cDNA Reverse Transcription Kit (Applied Biosystems, Foster, CA, USA). RNA extraction and reverse transcription were performed according to the manufacturer’s instructions with slight modifications [14].

### 2.5. Quantitative Real-Time PCR (qPCR)

The ZNF423 and matrix metalloproteinase 9 (MMP9) mRNA expression levels in cell lines were determined using a qPCR method. β-Actin was used as an internal control. cDNA was amplified using a gene specific TaqMan probes (Hs00323880_m1 ZNF423, Hs99999903_m1 β-actin and Hs00957562_m1 MMP9). qPCR was performed on an ABI Real-time PCR system, QuantStudio™ 6 Flex (Life technologies, Singapore). The cycle threshold (Ct) values were calculated using a delta Ct (2^−ΔCt^) method.

### 2.6. Western Blot Analysis

Protein extraction from cell pellets was performed using RIPA buffer (150 mM NaCl, 50 mM Tris-HCl, 1% (*v*/*v*) Triton X-100, 1% (*w*/*v*) Sodium deoxycholate, 0.1% (*w*/*v*) SDS and protease inhibitors cocktail). Then, protein concentration was determined using a Pierce™ BCA Protein assay kit (Pierce Biotechnology, Rockford, IL, USA). All of the extracted proteins from cell lysates were solubilized in SDS loading buffer. The protein solutions were boiled at 95 °C for 10 min and cooled on ice. An appropriate amount of protein was loaded onto an SDS-PAGE (4% stacking gel and 10% separating gel) and transferred onto a PVDF membrane (Merck, Billerica, MA, USA). The transferred membrane was blocked in 5% skim milk in Tris-buffered saline (TBS) pH 8.0 at room temperature for 1 h before being incubated with rabbit anti-ZNF423 antibody (1:500, Merck KGaA, Darmstadt, Germany), rabbit anti-MMP9 antibody (1:200, Cell Signaling Technology, Danvers, MA, USA), mouse anti-N-cadherin antibody (1:250, BD Biosciences, San Jose, CA, USA), rabbit anti-vimentin antibody (1:1000, Abcam, Cambridge, UK), mouse anti-ox-A1AT antibody (1:500, Ikagaku Co. Ltd., Kyoto, Japan) and mouse anti-β-actin antibody (1:25,000, Sigma, Louis, MO, USA) overnight at 4 °C. After incubation, the membranes were rinsed with TBS containing 0.1% Tween-20. Then, the membranes were incubated with peroxidase-labelled anti-rabbit or anti-mouse secondary antibodies for 1 h and rinsed with TBS containing 0.1% Tween 20. Finally, the membranes were exposed to Amersham™ ECL™ Prime Western Blotting Detection Reagent (GE Healthcare, Buckinghamshire, UK) for chemiluminescence detection which was detected using an Amersham Imager™ 600 (GE Healthcare Bio-Sciences AB, Uppsala, Sweden). In this study, β-actin was used as an internal control.

### 2.7. H_2_O_2_ Treatment of MMNK1 Cells

MMNK1 cells, which have low expression levels of ZNF423, were plated into 6-well plates at a seeding density of 1 × 10^5^ cells/well. Then, the cells were treated with 0, 25, 50 and 100 µM of H_2_O_2_ (Merck KGaA, Darmstadt, Germany) in Ham’s F-12 medium and incubated in a humidified atmosphere of 95% air and 5% CO_2_ at 37 °C for 24, 48 and 72 h. For 48 and 72 h of the H_2_O_2_ treatments, the cells were re-treated with H_2_O_2_ every 24 h. Subsequent to treatment with H_2_O_2_, the cells were washed with PBS and subjected to RNA extraction using TRIZOL^®^ reagent and protein extraction using RIPA buffer.

### 2.8. Knockdown of ZNF423 Using siRNA

The ZNF423-siRNA was designed by GE Healthcare Dharmacon Inc., Lafayette, CO, USA (ON-TARGETplus Human ZNF423 siRNAs; cat no. L-012907-00-0005). KKU-100 (9 × 10^4^ cells/well) and KKU-213 (12 × 10^4^ cells/well) were seeded into 6-well plates and transfected with 50 nM siRNA against ZNF423 according to the manufacturer’s protocol using lipofectamine^®^ RNAiMax (Invitrogen, Carlsbad, CA, USA). Cells transfected with non-targeting siRNAs (scramble) were used as negative controls. Then, the transfected cells were incubated for 72 h at 37 °C in an incubator with 5% CO_2_ and 95% relative humidity at 37 °C. ZNF423 expression levels, cell proliferation, colony forming ability, cell invasion and MMP9 expression levels were determined. 

### 2.9. Cell Proliferation Assay

Cell proliferation was determined using a Sulforhodamine B (SRB) assay. Cells were trypsinized and seeded at 2 × 10^3^ cells (KKU-100) and 3 × 10^3^ cells (KKU-213) per well into 96-well flat-bottom microtiter plates (quintuplicate per condition) and incubated for 24, 48 and 72 h at 37 °C in 95% humidified air with 5% CO_2_. Subsequently, 10% cold trichloroacetic acid (TCA) was added to the samples and incubated at 4 °C for 1 h. TCA-treated cells were stained with 0.4% (*w*/*v*) SRB in 1% (*v*/*v*) acetic acid for 45 min and washed three times with 1% (*v*/*v*) acetic acid to remove unbound dye. The plates were allowed to dry and the protein-bound SRB was solubilized with 200 μL of 10 mM Tris-base (pH 10.5) for 60 min on a shaker. The absorbance was measured at 540 nm using a microplate reader (Tecan Austria GmbH, Salzburg, Austria).

### 2.10. Clonogenic Assay

Cells having colony forming activity were enumerated using a clonogenic assay. Cells were trypsinized and seeded at 4 × 10^2^ cells (KKU-100) and 2 × 10^2^ cells (KKU-213) in 2 mL of Ham’s F-12 medium) into 6-well plates. Following incubation at 37 °C in 95% humidified air with 5% CO_2_, the medium was changed once every two days. After 10 or 14 days of incubation, the colonies were washed with 1 mL of PBS for 5 min and fixed with 1 mL of 4% (*w*/*v*) paraformaldehyde in PBS for 20 min. Thereafter, the fixed colonies were washed with 1 mL of PBS for 5 min and stained with 1 mL of 0.5% (*w*/*v*) crystal violet in distilled water for 10 min. The un-bounded dye was removed, and the plate was rinsed with tap water. Finally, the plate was dried, and the number of colonies were investigated under a microscope and counted using ImageJ free software.

### 2.11. Cell Invasion Assay

Cell invasion was determined using Corning^®^ BioCoat™ Matrigel^®^ Invasion Chamber (Discovery Labware, Inc., Bedford, MA, USA). The ZNF423-knocked-down KKU-100 (1.5 × 10^5^ cells) and KKU-213 (4 × 10^4^ cells) cells in the serum free medium were seeded onto the top of matrigel of the upper chambers and the complete medium containing 10% fetal bovine albumin was added into the lower chambers. The chambers were incubated at 37 °C in humidified air with 5% CO_2_. After 18 h of incubation, cells were allowed to invade through a porous 8 µm polyethylene terephthalate (PET) membrane toward the chemoattractant in the lower chamber and attached at the underside of the filter. The non-invading cells remaining on the upper surface of the membrane were removed by scrubbing with a sterile cotton swab. Then, the invaded cells were fixed with absolute methanol and stained with Mayer’s haematoxylin overnight. After staining, the membranes were washed with PBS and distilled water. The number of the invaded cells were counted under a light microscope.

### 2.12. Statistical Analysis

In this study, statistical analysis was performed using SPSS Statistic software version 17.0 (IBM Cooperation, Armonk, NY, USA). The survival curve was created and analyzed using Kaplan-Meier estimate with Log-rank test. The correlation between ZNF423 and 8-oxodG in all tissues were determined by Pearson’s correlation coefficients test and Chi-square test. Student *t*-test test was used for distinguishing between control and siZNF423 experiments. ANOVA test was used for the differentiation of ZNF423 mRNA expression levels in H_2_O_2_-treated cells. *p*-Value < 0.05 was considered to be statistically significant.

## 3. Results

### 3.1. ZNF423 Expression and 8-oxodG Detection in CCA Tissues

The expression levels of ZNF423 protein in CCA tissues (*n* = 75) were determined using IHC. The results showed that ZNF423 was over-expressed in CCA cells compared to the cholangiocytes in the adjacent non-cancerous areas (*p* < 0.001; graph not shown). Using a mean value of IHC score of CCA tissues as the cut-off point, 41% (31/75) of CCA tissues showed high ZNF423 expression (Figure 1A) and 53% (40/75) of CCA tissues showed high 8-oxodG formation (Figure 1B). The ZNF423 levels in CCA tissues were positively correlated to the 8-oxodG levels (*p* < 0.01) as shown in Table 1. In addition, as presented in a Kaplan–Meier plot (Figure 1C), when comparing survival between two groups of CCA patients, the group of patients having high levels of both ZNF423 and 8-oxodG had significantly (*p* = 0.047, log-rank test) shorter survival time compared with those patients with the other patterns (i.e., high ZNF423 alone, high 8-oxodG alone and low levels of both ZNF423 and 8-oxodG). These data suggested that the over-expression of ZNF423 in CCA is possibly induced by oxidative stress leading to CCA progression with poor prognosis.

### 3.2. Correlation between ZNF423 Expression Levels and Clinical Parameters of CCA Patients

All CCA tissues (*n* = 75) used in the present study were from intrahepatic CCA patients (52 males and 23 females). The median age of the patients was 57.8 years. Of 75 CCA patients, 42 (56%) have metastasis, whereas 33 (44%) have no metastasis. Histological types of CCA tissues were classified as the tubular type of 53% (40/75) and the papillary type of 47% (35/75). As shown in Table 1, none of those parameters were correlated with ZNF423 expression levels.

### 3.3. Effects of Oxidative Stress on ZNF423 Expression in an Immortalized Cholangiocyte Cell Line

To assess the effect of oxidative stress on the ZNF423 expression, MMNK1 (i.e., a low ZNF423 expressing immortalized cholangiocyte cell line) was treated with various concentrations of H_2_O_2_ (0, 25, 50 and 100 µM) for 24, 48 and 72 h. The results showed that ZNF423 mRNA expression (*n* = 2 per condition) was significantly increased in H_2_O_2_ treated cells at 48 and 72 h (Figure 2A). Moreover, ZNF423 protein levels and the oxidative stress marker (ox-A1AT) levels were increased in the H_2_O_2_ treatment as shown in Figure 2B. ox-A1AT can be used as an acute and chronic oxidative stress marker in CCA and the ox-A1AT levels were correlated with CCA pathogenesis [27,28]. The results given here strongly suggest that ZNF423 expression can be upregulated by H_2_O_2_, which is a commonly encountered ROS and oxidative stress.

### 3.4. Roles of ZNF423 in Proliferation and Invasion of CCA Cell Lines

To evaluate the roles of ZNF423 in CCA cells, two CCA cell lines, including KKU-100 and KKU-213, which were certified from the JCRB were selected for ZNF423-knockdown experiments. ZNF423 gene silencing with siRNA was found to effectively suppress mRNA and protein levels of ZNF423 compared with scrambled controls (Figure 3A,B) at 72 h after treatment. The cell proliferation rates of siZNF423-treated CCA cell lines were significantly reduced compared to control cells as shown in Figure 3C,D. Clonogenic assays showed less colonies in the ZNF423-knockdown group both in KKU-100 and KKU-213 cells than in scrambled controls (Figure 3E,F). Moreover, the siZNF423-transfected KKU-100 and KKU-213 cells were significantly reduced invasion ability compared to control cells (Figure 4A,B). In addition, MMP9 mRNA and protein expression levels were significantly decreased in siRNA-treated CCA cells compared to those in the control cells (Figure 4C,D). Moreover, the epithelial-mesenchymal transition (EMT) markers (N-cadherin and vimentin) were also decreased in the siRNA-treated cells compared to the control cells (Figure 4D). Our findings suggest that ZNF423 has an oncogenic property that can induce cell proliferation, colony forming ability and invasion in CCA via increased MMP9 expression and EMT processes. 

## 4. Discussion

Chronic oxidative stress causes DNA damage and increases the levels of DNA adducts, notably 8-oxodG, which is a well-known marker of oxidative stress and DNA damage [21,22,29]. Formation of 8-oxodG can be used to assess oxidative stress levels in biological samples. Oxidative stress has proved that it can induce the formation of 8-oxodG in CCA tissues and higher 8-oxodG formation was associated with poor prognosis of CCA patients [14,25]. Moreover, oxidative stress can also induce the alteration of gene expression via induction of genetic instability and epigenetic changes [25,30]. The present study showed that ZNF423 expression in MMNK1, was augmented after H_2_O_2_ treatment. H_2_O_2_ is a well-known oxidative stress inducing factor via an increase in superoxide anion (O_2_^•−^) generation through activation of NADPH oxidase [31,32,33]. Moreover, in this study, ZNF423 expression levels in CCA tissues were correlated with 8-oxodG formation. Notably, CCA patients who had high ZNF423 and 8-oxodG levels in the cancer tissues have significantly shorter survival. These findings indicate that ZNF423 is an oxidative stress responsive gene which plays significant roles in CCA progression.

In this study, the functions of ZNF423 in CCA were investigated using siRNA technique in CCA cell lines. The results showed that ZNF423 is involved in the proliferation and colony forming ability of CCA cells. Related to this, ZNF423 has been shown to induce proliferation and colony formation of CML cell line permitting CML progresses into its late severe stages [9]. In addition, ZNF423-knockdown significantly reduced the invasion of CCA cells compared to the control cells. Thus, ZNF423 plays important roles in CCA progression via an increase in cell proliferation and invasion rates upon oxidative stress.

MMP9, also known as type IV collagenase or gelatinize B, is a Zn^2+^ endopeptidase that degrades extracellular matrix (ECM) proteins such as gelatin and collagen resulting in cancer metastasis and invasion [34,35]. MMP9 expression was associated with the World Health Organization (WHO) grade of glioma and induced glioma cell proliferation [36]. Also, MMP9 was involved in cell invasion of many cancers including CCA [37,38,39]. MMP9 expression was correlated with 8-oxodG and inducible nitric oxide synthase (iNOS) expression in an Ov-infection-induced CCA model, suggesting a link between MMP9 and oxidative stress [40]. Moreover, MMP9 is one of EMT regulators that plays significant roles in cancer invasion and metastasis [41]. These data suggest that MMP9 is a key regulator of cancer cell proliferation and invasion. 

In this study, the suppression of ZNF423 in CCA cells resulted in the decrease of MMP9 mRNA and protein expression levels. The suppression of ZNF423 also reduced the expression levels of EMT markers (N-cadherin and vimentin). Reduced expressions of N-cadherin and vimentin were found to be associated with EMT process, CCA progression and aggressive clinical outcomes such as poor overall survival and advanced-stage tumor [42]. Related to this, MMP9 expression was reported to be enhanced by retinoic acid (RA) through RARα and RARγ [43,44]. Moreover, ZNF423 was previously reported to act as a co-activator of RARα/RXRα heterodimers in neuroblastoma cells [11]. RARγ and RXRα were reported to play crucial roles in CCA cell proliferation [16,17]. Taking all these together, oxidative stress induces ZNF423 expression that might interact with RARs/RXRs and subsequently promote the proliferation and invasion of CCA cells by up-regulating MMP9 expression levels.

## 5. Conclusions

The present results show that *ZNF423* is an oxidative stress related gene which plays a pivotal role in driving CCA genesis. ZNF423 and its related proteins might be novel targets for CCA chemotherapy towards a further increase of survival rates and hereby improve the cancer patients’ quality of life.

## Figures and Tables

**Figure 1 biomolecules-09-00263-f001:**
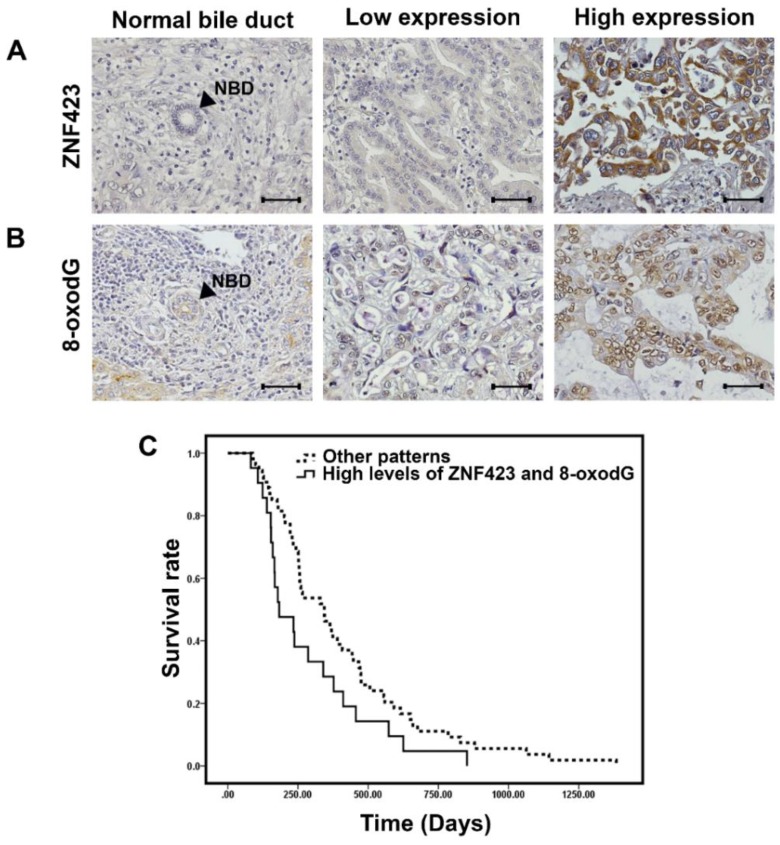
Representative immunohistochemical staining patterns of Zinc finger protein 423 (ZNF423) (**A**) and 8-oxodG (**B**) in cholangiocarcinoma (CCA) and a normal bile duct (NBD) in the adjacent noncancerous area. NBD was indicated by the arrows. All figures are 400x original magnification. Scale bar is equal to 50 µm. (**C**) Kaplan-Meier survival curves illustrating the correlation between the high levels of ZNF423 and elevated formation of 8-oxodG and overall survival rates in CCA patients. CCA patients with high ZNF423 expression adjunct with high 8-oxodG formation (*n* = 22) were significantly associated with short survival compared to other patterns (*n* = 53), corresponding to those patients with either high ZNF423 alone or 8-oxodG alone or low levels of both ZNF423 and 8-oxodG. Log-rank test analyses revealed significant difference between the two groups (*p* = 0.047).

**Figure 2 biomolecules-09-00263-f002:**
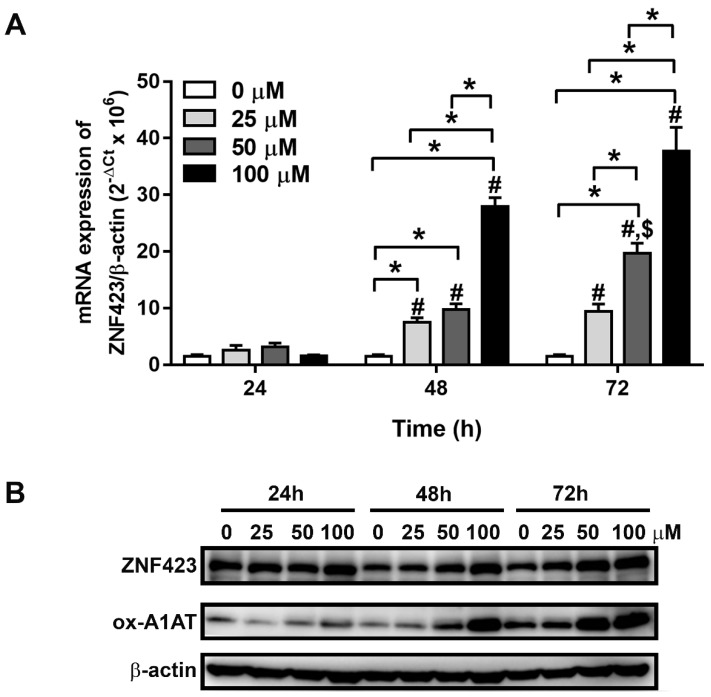
(**A**) ZNF423 mRNA expression levels and (**B**) protein expressions of ZNF423 (~100 kDa) and ox-A1AT (~52 kDa) in an immortalized cholangiocyte cell line (MMNK1) cells after treatment with various concentrations of H_2_O_2_ (0, 25, 50 and 100 µM for 24, 48 and 72 h). Significance was calculated by ANOVA test (* represents *p* < 0.05, # represents *p* < 0.05 compared with 24 h and $ represents *p* < 0.05 compared with 48 h). The mRNA expression levels of ZNF423 were normalized by β-actin.

**Figure 3 biomolecules-09-00263-f003:**
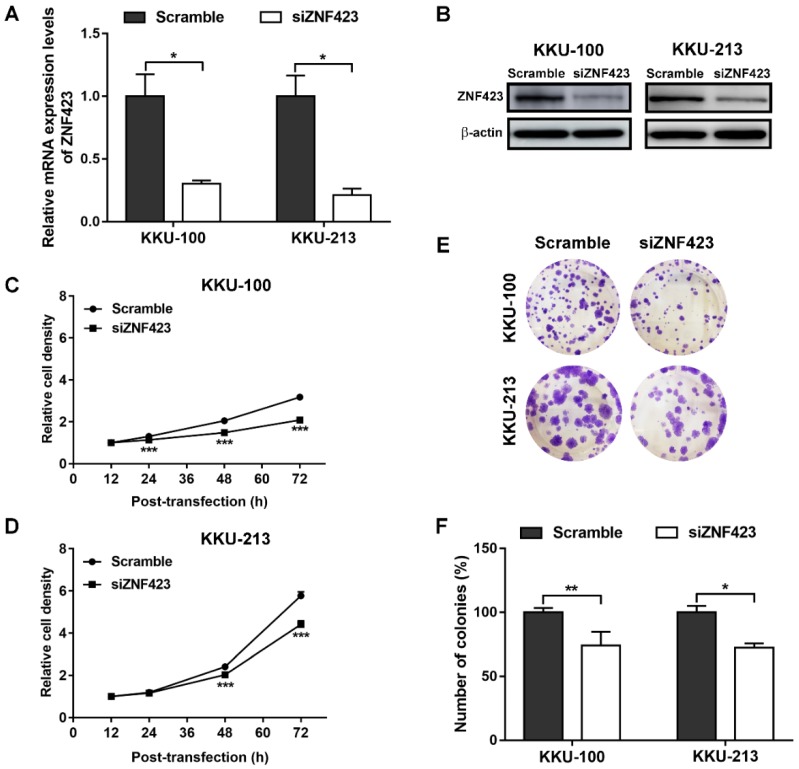
Effects of ZNF423 silencing on CCA cell proliferation and invasion. ZNF423 mRNA ((**A**); *n* = 2 per condition) and protein (~100 kDa) (**B**) Expression levels, in KKU-100 and KKU-213, of siRNA-treated and scramble-treated KKU-100 and KKU-213 cells were measured using Quantitative Real-Time PCR (qPCR) and western blot analysis and normalized by β-actin (~42 kDa) expression levels. (**C**,**D**; *n* = 5 per condition) Relative cell density analyzed by SRB assay in KKU-100 and KKU-213 cells. (**E**) The crystal violet-stained KKU-100 and KKU-213 cells after cell re-seeding for 10 and 14 days. (**F**; *n* = 2 per condition) Relative number of colonies (%) in KKU-100 and KKU-213 cell lines. *P*-values were calculated by Student’s *t* test compared with scramble conditions (* *p* < 0.05, ** *p* < 0.01, and *** *p* < 0.001).

**Figure 4 biomolecules-09-00263-f004:**
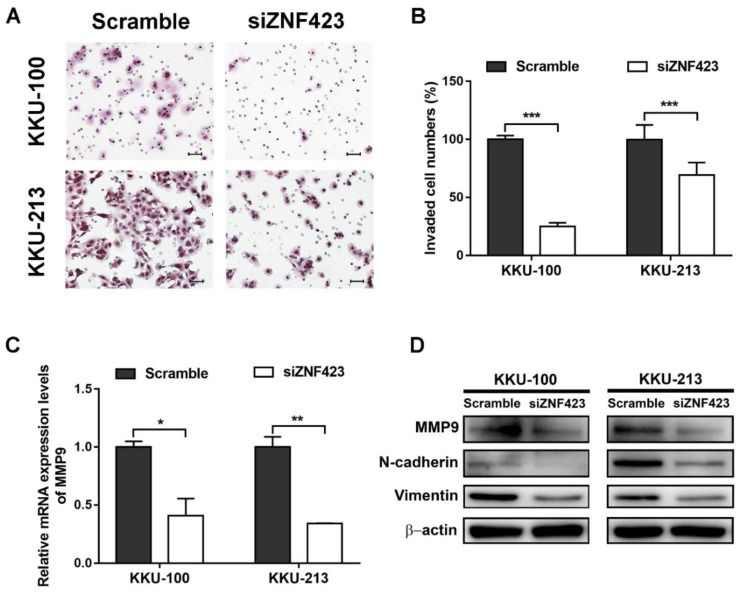
(**A**) Invasion of CCA KKU-100 and KKU-213 cells transfected with siRNA targeting ZNF423 and non-targeting scrambled controls. Scale bar is equal to 50 µm. (**B**) Representative percentage of invaded cells in KKU-100 and KKU-213 (*n* = 2 per condition). (**C**) MMP9 mRNA expression levels of siRNA-treated and scramble-treated KKU-100 and KKU-213 cells were determined using qPCR (*n* = 2 per condition). (**D**) Western blot analysis of MMP9 (~92 kDa), N-cadherin (~130 kDa), vimentin (~54 kDa) and β-actin (~42 kDa) protein expression levels. *P*-values were calculated by Student’s *t* test compared with scramble conditions (* *p* < 0.05, ** *p* < 0.01, and *** *p* < 0.001).

**Table 1 biomolecules-09-00263-t001:** Correlation between ZNF423 expression levels in cholangiocarcinoma (CCA) tissues and clinico-pathological data of CCA patients and 8-oxodG formation.

Variable	ZNF423		*p*-Value
	Low	High	
Survival day (Median (min-max))	297.5 (89-1384)	260 (82-852)	0.101
Metastasis -Non-metastasis -Metastasis	20 24	13 18	0.474
Histology -Tubular -Papillary	24 20	16 15	0.493
Gender -Male -Female	29 15	23 8	0.306
Age ≤57.8 ≥57.8	23 21	13 18	0.259
Age (Mean ± SD)	56.97 ± 7.72	59.03 ± 6.81	0.228
8-oxodG -Low -High	26 18	9 22	0.009

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
