# Peer review of "Roles of Zinc Finger Protein 423 in Proliferation and Invasion of Cholangiocarcinoma through Oxidative Stress"

_biomolecules, 2019, doi:10.3390/biom9070263_

Round 1

Reviewer 1 Report

The authors showed that ZNF423 is an oxidative stress related gene which plays a pivotal role in driving cholangiocarcinoma genesis. Overall the study appears well done and the data are clearly presented. However, this topic is not suitable for Biomolecules. Because this journal focuses on biogenic substances and their biological functions, structures, interactions with other molecules, and their microenvironment as well as biological systems.

Minor points:

1. Line 44-45: Connect two sentences or rewrite

2. Table 1. Correlation…

3. Statistical analysis (Fig. 2, Fig. 3): The t-test compares the means between 2 samples, but if there is more than 2 conditions in an experiment a ANOVA is required.

4. Fig. 2, Fig. 3A, C, D, E, Fig. 4B, C: Please show a group size (n) that subjected to statistical analysis.

5. The discussion section is a little short.

Author Response

Reviewer#1

The authors showed that ZNF423 is an oxidative stress related gene which plays a pivotal role in driving cholangiocarcinoma genesis. Overall the study appears well done and the data are clearly presented. However, this topic is not suitable for Biomolecules. Because this journal focuses on biogenic substances and their biological functions, structures, interactions with other molecules, and their microenvironment as well as biological systems.

Reply to the comment: Thank you very much for your supportive comments. We changed the topic to “Roles of Zinc finger protein 423 in proliferation and invasion of cholangiocarcinoma through oxidative stress”. We hope this topic is suitable for Biomolecules.

Minor points:

1.      Line 44-45: Connect two sentences or rewrite

Reply to the comment#1: Thank you for your suggestion. The two sentences have been rewritten.

2.      Table 1. Correlation…

Reply to the comment#2: Thank you very much for your suggestion. We have changed “correlation” to “Correlation”.

3.      Statistical analysis (Fig. 2, Fig. 3): The t-test compares the means between 2 samples, but if there is more than 2 conditions in an experiment and ANOVA is required.

Reply to the comment#3: Thank you very much for your comment. In the revised manuscript, ANOVA has been used for our statistical analysis. In the revised manuscript, the word “t-test” has been changed to “ANOVA”.

4.      Fig. 2, Fig. 3A, C, D, E, Fig. 4B, C: Please show a group size (n) that subjected to statistical analysis.

Reply to the comment#4: Thank you very much for your careful review. We added the information on the group size for each statistical analysis.

5.      The discussion section is a little short.

Reply to the comment#5: Thank you very much for your valuable comment. We discussed more and expanded our discussion section.  The references no. 38-40 have also been added to support our work in the revised manuscript.

Reviewer 2 Report

This manuscript showed that ZNF423 is an oxidative stress responsive gene during CCA progression in vivo and in vitro.  However, I have questions for this conclusion. I propose that to improve the manuscript and to increase potential impact figures in this field of scientific research.

Major comments

Comment 1

Author write “in a dose- and time-dependent manner~~” in line236-238. Therefore, for statistical analysis, it is best to do a statistical test using two-way analysis (two factors; time and dose). Student’s t test is not suitable for your conclusion. This point is very important.

Comment 2

In line239, author indicated that ZNF423 expression can be upregulated by H2O2, which is a commonly encountered ROS. I think that author should be measure the accumulating ROS production at various condition in Fig.2 and Fig.3 (in ZNF423 knockdown KKU-100 and KKU213). Furthermore, author should check cell viability and proliferation. As it is, I think that it’s really an unreasonable conclusion.

Comment 3

Author measured RNA expression in figure 2 (ZNF423) and 4C (MMP9). However, it is not enough. Thus, author need to evaluate expressions level of protein using western blotting. In addition, to assess the cell invasion, author should check the MMP2 protein level. This point is very important.

Comment 4

EMT is a crucial process in cancer progression that promotes cancer cells. Thus, author must measure Snail, Vimentin and N-cadherin under ZNF423 knockdown cells compare to siControl cells.

Round 2

Reviewer 1 Report

The authors have responded to all my questions and made the necessary changes to the manuscript.

Author Response

Thank you very much for your supportive comments.

Reviewer 2 Report

This manuscript is improved. However, this manuscript must improve in this field of scientific research.

Major comments

Comment 1

Actually, did author perform the two-way anove in Fig.2? You wrote “in a dose- and time-dependent manner~~” in line236-238. Because, you indicate 0uM is 1 (relative value) in all time. Again, for statistical analysis, it is best to do a statistical test using two-way analysis (two factors; time and dose). This point is very important. As it is, I think that it’s really an unreasonable conclusion. As well as Figure R1.

Author Response

Please look at the attached file.

Round 3

Reviewer 2 Report

I think accept this re-manuscript.

However, author should more study and understrand about statistics analysis.